# Removal of Oily Contaminants from Water by Using the Hydrophobic Ag Nanoparticles Incorporated Dopamine Modified Cellulose Foam

**DOI:** 10.3390/polym13183163

**Published:** 2021-09-18

**Authors:** Nadeem Baig, Irshad Kammakakam

**Affiliations:** 1Interdisciplinary Research Center for Membranes and Water Security, King Fahd University of Petroleum and Minerals, Dhahran 31261, Saudi Arabia; 2Department of Chemical & Biological Engineering, University of Alabama, Tuscaloosa, AL 35487-0203, USA

**Keywords:** polymers, separation, nanoparticles, oil pollution, water, environment

## Abstract

The presence of oil-related contaminants in water has emerged as a severe threat to the environment. The separation of these contaminants from water has become a great challenge, and extensive efforts are being made to develop suitable, environmentally friendly materials. Highly hydrophobic materials are effective in the selective separation of oil from water. In this work, silver (Ag)-incorporated, highly hydrophobic dopamine-modified cellulose sponge was prepared by functionalizing with the range of alkyl silanes. The Ag nanoparticle-incorporated dopamine provided the appropriate roughness, whereas the alkyl component provided the low surface energy that made it selective towards oil. It was found that the alkyl groups with a longer chain length were more effective in enhancing the hydrophobicity of the Ag nanoparticle-incorporated, dopamine-modified cellulose. The developed materials were characterized by Fourier transform infrared spectroscopy (FTIR), field emission-scanning electron microscopy (FE-SEM), energy-dispersive X-ray spectroscopy (EDX), elemental mapping, and contact angle goniometry. The maximum water contact angle on the functionalized surfaces was observed at 148.4°. The surface of the C18s-Ag-DA-Cell-F showed excellent selectivity towards the oily component that rapidly permeated, and water was rejected wholly. The developed material showed a separation efficiency of 96.2% for the oil/water mixture. The C18s-Ag-DA-Cell-F material showed excellent reusability. Due to their environmentally friendly nature, excellent selectivity, and good separation efficiency, the functionalized cellulose materials can be used to separate oil and water effectively.

## 1. Introduction

The separation of oil-related organic contaminants from water has emerged as a critical environmental problem. The significant contributors to oil-related contaminants in water are oily discharge from industry [1], occasional oil spills, and household waste. The risk of oil spills is increasing due to the increasing marine oil exploration and the rapid offshore movement of oil. Oil spills can cause severe health and environmental impacts and enormous economic losses [1]. Oil spills in water may cause oxygen deficiency in aquatic systems, leading to the death of living organisms [2]. The separation of oil-related contaminants from water has received significant importance due to their substantial adverse impact on the economy and the environment. It has become an area of great interest to develop materials and methods to remove these contaminants from water with high efficiency [3,4].

The specific wettable surfaces proved effective for the separation of oil from water [5]. The surface wettability term defines the interaction between the liquid and the solid phases [6]. Nature provides several examples of special wettable surfaces such as lotus leaves [7] and fish scales [8]. The materials with special wettability generally operate by allowing one phase to infiltrate while rejecting the opposite one [9]. The surfaces famous for separating oil and water usually appear as either (a) hydrophilic/oleophobic or (b) oleophilic/hydrophobic in nature. Hydrophilic and oleophobic surfaces show an affinity towards water [10], whereas oleophilic and hydrophobic surfaces show an affinity towards oil, rejecting water [11]. The switchable materials are also being used to separate oil from water efficiently [12].

In recent years, great attention has been given to the fabrications of highly hydrophobic materials to separate oil and water [13]. Hydrophobic materials are used to remove oil from water, selectively. Various hydrophobic materials reported in the literature for the separation of oil and water include, but are not limited to, aerogels [14], xerogels [15], brass filters [16], meshes [17], sponges [18], woven fabrics [19], filter papers [20], porous ceramics [21] and membranes [22]. Several methods have been used to produce superhydrophobic materials, such as dip coating [23], spray coating methods [24], chemical vapor deposition [25], hydrothermal methods [26], and layer-by-layer assembly [27].

The literature has revealed that the interest in compounding the inorganic and organic moieties to achieve highly effective oil and water separation has been increasing [28]. There are several examples in the literature in which the combination of organic moieties and inorganic materials have been used to produce the highly hydrophobic surfaces for oil and water separation. The Maiping Yang group [29] produced fluorine-free superhydrophobic surfaces by combining polydimethylsiloxane (PDMS) and titanium dioxide (TiO_2_). In another work, TiO_2_ was deposited on the fabrics by a hydrothermal method and, later, functionalized with the 1H,1H,2H,2H-perfluorodecyltriethoxysilane to produce a low-energy surface for oil and water separation [30]. A superhydrophobic absorber was developed to effectively collect oil by painting SiO_2_ particles/polythiophene on cotton fabrics [31]. The hydrophobic ZnO was synthesized by autoclaving zinc chloride and treating the ZnO at room temperature with stearic acid [32]. These efforts were continued in order to produce the best material with a combination of inorganic and organic materials to effectively separate oil and water.

Cellulosic polymer materials are considered superior in oil/water separation due to their biodegradability, low cost, and abundance in nature [20]. Cellulose is inherently hydrophilic in nature [33], and chemical modification is required to achieve the hydrophobic surface [34]. To fabricate highly hydrophobic surfaces, generally, rough surfaces are required. The rough surfaces can be produced by etching or coating with various micro- or nanoparticles [35,36]. The functionalization of rough surfaces with low-energy materials results in hydrophobic surfaces, showing the incredible ability to resist water penetration. In this work, the highly hydrophobic cellulose sponge is developed to separate oil and water. An Ag-PDA layer is produced on the surface of the cellulose. The Ag nanoparticles are introduced to produce the roughness on the surface of the cellulose foam and polymerized dopamine provides the stability to the surface. Then, the Ag-PDA modified cellulose foam is functionalized with the range of Cn-silanes to find the optimum hydrophobic cellulose for the effective separation of oil and water. The developed C18-Ag-DA-Cell-F shows excellent performance in the separation of oily contaminants from water.

## 2. Materials and Methods

### 2.1. Materials

AgNO_3_ (purity > 99%), Triethoxyoctylsilane (purity ≥ 97.5%), Isobutyl (trimethoxy)silane (purity 97%), Octadecyltrichlorosilane (≥90%) and Trimethoxymethylsilane (purity 98%) was purchased from Sigma Aldrich, St. Louis, MO, USA. Ethanol (99.8%) was received from Honeywell Riedel-de-Haën™, Seelze, Germany. Cellulose foam was collected from the local market. Toluene (purity 99.8%) was received from ALPHA CHEMIKA, Mumbai, India. Methylene blue (purity 90%) was obtained from BDH Chemicals Ltd., Poole, England. Dichloromethane (purity 99.5%) was purchased from the EM Science, Darmstadt, Germany. The deionized water was used to conduct the various experiments and to prepare the various required solutions. The deionized water was collected from the PURELAB flex with a resistance of 18.2 MΩ.

### 2.2. Instrumentation

Various characterization tools were used to characterize the functionalized materials. Functional groups on the surface of the Cell-F were established by scanning the FTIR of the materials with the Thermo Scientific Nicolet iS10 spectrometer. The lens surface of the Thermo Scientific Nicolet iS10 spectrometers was cleaned with isopropanol. The solid samples were directly placed on the sample holder space for recording FTIR. The samples were scanned over the range of 400 to 4000 cm^−1^. Each sample result was collected after 60 scans at a spectral resolution of 32 cm^−1^, and peak repeatability was confirmed by repeating the samples two times at different places. The Goniometer DSA25 KRÜSS (Hamburg, Germany) was used to measure the water contact angle on the surface of the various non-functionalized and functionalized Cell-F. The contact angles were recorded by placing a 5 µL drop of deionized water on the material surfaces. Before measuring the contact angle, the stabilization time of 15 s was given before measurements were started. A Thermo Scientific™ Quattro field-emission scanning electron microscope was used to study the surface morphology. Before scanning, the samples were coated with gold by sputtering of the Au sources. All SEM images were recorded in the secondary electron mode at the accelerating voltage of 20 kV. The Samsung camera was used to capture the various images of the materials used throughout the experiment.

### 2.3. Development of Functionalized Cellulose Foam

The Cell-F was cut into appropriate pieces, cleaned well with deionized water, and dried. The 0.01 M solution of the dopamine was prepared and adjusted pH of about 8.5. The pH of the solution was adjusted with the help of the dilute NaOH solution. The dried Cell-F was dipped into the dopamine solution and placed in the water shaker bath at room temperature. After 3 h, the DA-Cell-F was taken out and dried in the oven at 50 °C. The 0.1 M solution of AgNO_3_ was prepared in deionized water. The DA-Cell-F dipped into the 0.1 M solution of AgNO_3_ and was kept on the water shaker bath for better interaction of the DA-Cell-F surface and the AgNO_3_ solution for 4 h. After 4 h, the Ag-DA-Cell-F was taken out and dried at 50 °C. A 2% solution of Trimethoxymethylsilane (C1s), Isobutyl (trimethoxy)silane (C4s), Triethoxyoctylsilane (C8s), and Octadecyltrichlorosilane (C18s) were prepared in the toluene. The solutions of the various silanes were sonicated for 15 min. The Ag-DA-Cell-Fs were kept in the 2% solution of the Cns for 3 h. The various functionalized Cell-F (C1s-Ag-DA-Cell-F, C4s-Ag-DA-Cell-F, C8s-Ag-DA-Cell-F, and C18s-Ag-DA-Cell-F) were washed with ethanol to remove the excess and unreacted Cns. After washing, the functionalized Cell-Fs were dried and preserved for further characterization and experimental evaluations for the oil/water separation. The schematic illustration of the fabrication of the various functionalized foam can be seen in Figure 1.

### 2.4. Functionalized Cellulose Foam for Oil/Water Separation

For oil/water separation, highly hydrophobic C18s-Ag-DA-Cell-F was fitted into the separation assembly by placing it between the funnel and base containing the PFTF support. The funnel and base were kept tight with the help of the anodized aluminum clamp. The whole separation setup was adjusted into the 400 mL Pyrex glass receiving beaker. The oil/water mixture was prepared by taking 50 mL of dichloromethane and 50 mL of water. The coloring agent in the water was methylene blue to make a sharp distinction from the colorless, oily component. The oil/water mixture was slowly poured into the separation assembly. The water was repelled by C18s-Ag-DA-Cell-F and stayed in the funnel; however, the oil passed through the C18s-Ag-DA-Cell-F into the collector beaker.

## 3. Results and Discussion

### 3.1. FTIR

FTIR spectroscopy proved a powerful tool in determining the successful functionalization of the various surfaces over time. From the cellulose structure, it is clear that the cellulose is full of hydroxyl groups. The FTIR spectrum showed the strong presence of the hydroxyl groups at 3355 cm^−1^ (Figure 1a). Due to the hydrophilicity of the cellulose, some of the water adsorbed on the surface of the cellulose; therefore, the HOH bending band appears at 1642 cm^−1^, and this value is close to the reported HOH bending band for bulk water [37,38]. The absorption band that appears at 1031 cm^−1^ is attributed to the C-O stretching. The absorption band that is observed at 668 cm^−1^ is assigned to the -OH out of plane bending vibrations [39,40]. After introducing the various alkyl silanes, a dramatic change in the FTIR spectra of the Cell-F can be observed. The intensity of the -OH absorption band at 3355 cm^−1^ is substantially reduced, and this is observed in the case of all silanes interacting with Cell-F such as C1s-Ag-DA-Cell-F, C4s-Ag-DA-Cell-F, C8s-Ag-DA-Cell-F, and C18s-Ag-DA-Cell-F (Figure 1).

The sharp reduction in the -OH absorption band′s intensity proves that most of the hydroxyl groups interacted with the various silanes. Furthermore, the H-OH bending vibrations after interaction with silanes almost disappear. It seems that moisture adsorption on the Cell-F is discouraged after functionalization with various silanes. The absorbance band at 2977 cm^−1^ appears due to the -C-H stretching of the terminal -CH_3_ groups, and it appears more prominent in the case of C1s-Ag-DA-Cell-F and C4s-Ag-DA-Cell-F compared to the C8s-Ag-DA-Cell-F and C18s-Ag-DA-Cell-F. In the case of C8s-Ag-DA-Cell-F and C18s-Ag-DA-Cell-F, the -CH_2_ stretching vibration absorption band dominantly appears at 2924 cm^−1^ and 2917 cm^−1^, respectively. The absorption band appearing at 1089 cm^−1^ and 1086 cm^−1^ is assigned to the Si-O vibrations in the C8s-Ag-DA-Cell-F and C18s-Ag-DA-Cell-F, respectively [41]. The sharp reduction in the hydroxyl absorption band of the Cell-F and DA-Cell-F after introducing the silanes and appearance of the Si-O vibrations shows the successful functionalization of the Cell-F with the various silanes.

### 3.2. Morphological and Elemental Characterization of the Functionalized Cell-F

The surface morphology of the Cell-F, DA-Cell-F, Ag-DA-Cell-F, C1s-Ag-DA-Cell-F, C4s-Ag-DA-Cell-F, C8s-Ag-DA-Cell-F and C18s-Ag-DA-Cell-F is characterized with the help of field emission-scanning electron microscopy. The Cell-F surface in Figure 2A appears relatively smooth at higher magnification. After modification with dopamine, some layers appear on the surface of the DA-Cell-F. The formation of polydopamine is also evident from the photograph, in which sharp color changes can be observed after the formation of the polydopamine layer. The SEM images of the Ag-DA-Cell-F, C1s-Ag-DA-Cell-F, and C4s-Ag-DA-Cell-F can be seen in Appendix A. In elemental mapping, the nitrogen also appears that is absent in the case of the Cell-F. In Ag-DA-Cell-F mapping, the Ag also appears in addition to the C, O, and N (Appendix A). Polydopamine contributes the N in the mapping of the Ag-DA-Cell-F. Polydopamine is helpful in the incorporation of the various metal and metal oxide nanoparticles on the surface of the various materials [42]. In the case of C1s-Ag-DA-Cell-F, C4s-Ag-DA-Cell-F, C8s-Ag-DA-Cell-F, and C18s-Ag-DA-Cell-F, the presence of Ag is seen in the SEM images that result in the formation of the rough, hierarchal structure with polydopamine. It is also seen in the SEM images that on the rough surface of the Ag-DA-Cell-F, some rough extensions can be seen after interaction with the C8s and the C18s (Figure 2C,D). The rough extension might be produced after interaction with the various alkyl silanes. In elemental mapping of the C18s-Ag-DA-Cell-F (Figure 3), C, O, N, Ag, and Si is observed, and these elements appear after functionalization of the Cell-F. The silicone is contributed from the silane. The morphological changes and elemental mapping analysis provide the successful functionalization of the Cell-F.

### 3.3. Surface Wettability Analysis

The surface wettability behavior of the dopamine-modified Cell-F is evaluated after each modification step to find out the suitability for the selective separation of the oil-related contaminants from the water. The Cell-F and DA-Cell-F show a strong affinity for water, and their surfaces appear strongly hydrophilic in nature. The water drop immediately penetrates the foam as it comes in contact with them. Similar behavior is observed for C1s-Ag-DA-Cell-F and C4s-Ag-DA-Cell-F. It is found that the modification of the Ag-DA-Cell-F with the C1s and C4s alone is insufficient to prevent water penetration into the Ag-DA-Cell-F when water drops come in direct contact in air with C1s-Ag-DA-Cell-F and C4s-Ag-DA-Cell-F. A big blue mark is observed on the surface of the C1s-Ag-DA-Cell-F and C4s-Ag-DA-Cell-F in Figure 4C,D. The blue marks appeared due to the penetration of the methylene blue-colored water into C1s-Ag-DA-Cell-F and C4s-Ag-DA-Cell-F. It is observed that water slowly penetrates the C4s-Ag-DA-Cell-F compared to the C1s-Ag-DA-Cell-F. Modifying the Ag-DA-Cell-F surface with the C8s improves the surface hydrophobicity, and the surface shows some resistance to water drops. The surface of the Ag-DA-Cell-F, after functionalization with the C18s, has to become stably hydrophobic, and water drops remain circular on the surface of the C18s-Ag-DA-Cell-F. No blue stains are observed after the removal of water drops from the surface of the C18s-Ag-DA-Cell-F. Furthermore, the water drops freely move on the surface of the C18s-Ag-DA-Cell-F, which is not sticky anymore. Alkyl chain length has a positive effect on the hydrophobicity of the materials. The longer alkyl chain provides a better shield from the water than the smaller alkyl chains [43].

In-depth analysis of the surface hydrophobicity is further confirmed by measuring the water contact angle on the surface of the various Cell-F functionalized materials. According to the wettability criteria, the surfaces are hydrophilic if the static contact angle is less than 90°, and hydrophobic if the water static contact angle is greater than 90°. On the surfaces of the Cell-F, DA-Cell-F, C1s-Ag-DA-Cell-F, and C4s-Ag-DA-Cell-F, the water drops penetrate quickly; that is why zero water contact angle is observed (Appendix A). The water contact angle on the surface of the C8s-Ag-DA-Cell-F is found at the first stage at about 118°. On the surface of the C18s-Ag-DA-Cell-F, the WCA is observed at 148.4° (Figure 5). The uneven surfaces may cause some change in the contact angle. It is essential to note that it is difficult for the flat surface′s contact angle to exceed 120° even after introducing the low surface energy materials [30]. It is critical to introduce surface roughness, which helps to achieve highly hydrophobic surfaces. The Ag nanoparticles are incorporated into cellulose foam with the help of polydopamine to enhance the surface roughness. This is the reason for the high water contact angle of 148.4° that is observed on the surface of the C18s-Ag-DA-Cell-F. The highly hydrophobic surface of the C18s-Ag-DA-Cell-F is the combined effect of the surface roughness and low surface energy materials.

The water contact angle study indicates that the alkyl length of the silanes has a significant impact on the wettability of the Ag-DA-Cell-F. The alkyl silane interacts with the surface through the hydroxyl group. Some of the hydroxyl groups remain unreacted, which is observed in the FTIR spectra of the various Cns-Ag-DA-Cell-F. In the C1s-Ag-DA-Cell-F and C4s-Ag-DA-Cell-F, the water drops might make easy contact with the residual hydroxyl group, which helps them to penetrate the C1s-Ag-DA-Cell-F and C4s-Ag-DA-Cell-F. However, the penetration is a little bit slower compared to the C1s-Ag-DA-Cell-F. The surface shielding effect becomes more pronounced in the case of the long-chain alkyl functionalized C18s-Ag-DA-Cell-F. The water does not directly contact the residual hydroxyl groups, resulting in a highly hydrophobic water resistance surface. The functionalization of the hydrophilic surface, such as DA-Cell-F, with the alkyl chain containing fewer carbons atoms is not enough to turn the surface hydrophobic to such an extent where it can reject the water without leaving any watermark. The increasing chain length results in enhanced hydrophobicity [44].

### 3.4. Separation of the Oil/Water Mixture

The C18s-Ag-DA-Cell-F was used to separate the oil/water mixture due to its high hydrophobicity among the other Cn-Ag-DA-Cell-F and excellent capability to reject the water from its surface. The separation performance of the developed C18s-Ag-DA-Cell-F was evaluated by pouring a mixture of oil and water. A representative model of oil/water was developed with the help of dichloromethane (heavy oil) and water [45]. The water was colored blue by adding the methylene blue to make it recognizable from the oil component.

The C18s-Ag-DA-Cell-F was adjusted into the filtration setup (Figure 6A). The oil/water mixture was slowly poured into the separation assembly setup. The surface showed a selective response towards oily components and water. During the separation process, the surface rejected the water, and it kept collecting on the upper surface of the C18s-Ag-DA-Cell-F (Figure 6C). However, the behavior of the surface was found wholly opposite for the oily component. The oily component penetrated the spongy network of the C18s-Ag-DA-Cell-F and permeated to the collecting component of the separation setup. As can be seen in Figure 6D, the methylene blue-colored water stayed in the funnel part, whereas the oily component moved through the C18s-Ag-DA-Cell-F. The separation process was continued until the water and oily components completely separated from each other. The separation process of the oil/water mixture can be seen in Appendix A.

The developed C18s-Ag-DA-Cell-F showed significant separation efficiencies of the oil and water mixture. The separation efficiencies of the C18s-Ag-DA-Cell-F were calculated by using the following equation [24]:Separation efficiency (E%): M_1_/M_2_ × 100(1)

M_2_ is the weight of the oil in the mixture before separation, whereas M_1_ is the weight of the oil after separation. Before performing the separation efficiency analysis, the hydrophobic C18s-Ag-DA-Cell-F was saturated with the oily component to avoid oil absorption during the separation process. Upon pouring the oil/water mixture, the oil permeated while the water was rejected. The permeated oil was collected, and its weight was measured for the evaluation of the separation efficiency. The maximum separation efficiency was achieved at 96.2%, and the average separation efficiencies of the ten cycles of reusability were found at 94.0 ± 1.3% (Figure 7).

## 4. Conclusions

In this work, we successfully fabricated a range of Ag nanoparticle-incorporated dopamine-modified cellulose foams. The Ag nanoparticles provided roughness, whereas the dopamine provided stability on the surface of the cellulose foam. The surface selectivity of the Ag nanoparticle-incorporated dopamine-modified cellulose was investigated by functionalization with the alkyl groups. Functionalized surfaces were characterized with advanced instrumentation. It was found that the functionalized cellulose foam C1s-Ag-DA-Cell-F and C4s-Ag-DA-Cell-F were not effective in rejecting water. The surface of the C18s-Ag-DA-Cell-F was observed as suitable for separating oil and water due to its remarkable selectivity for oil in the mixture of oil and water. When the surface of the C18s-Ag-DA-Cell-F was exposed to the mixture of oil and water, the oil selectively permeated, whereas the surface rejected the water. The separated oil was found to be colorless, and no methylene blue-colored water was observed in the separated oil. The surface of the C18s-Ag-DA-Cell-F appeared highly hydrophobic, with a water contact angle greater than 148°. The maximum separation efficiency of the C18s-Ag-DA-Cell-F was found to be about 96.2%. Among various functionalized cellulose materials, the C18s-Ag-DA-Cell-F was suitable for separating oil from water due to its high hydrophobicity, excellent separation efficiency, and reusability. Due to the extensive natural availability of cellulose and the environmentally friendly nature of the material, the synthesized materials can be produced at a large scale to clean spilled oil from water.

## Data Availability

Not applicable.

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
