# Peer review of "Removal of Oily Contaminants from Water by Using the Hydrophobic Ag Nanoparticles Incorporated Dopamine Modified Cellulose Foam"

_polymers, 2021, doi:10.3390/polym13183163_

Round 1
Reviewer 1 Report
The manuscript “Removal of oily contaminants from water by using the hydrophobic nanoparticle incorporated dopamine modified cellulose foam” by Baig and Kammakakam describes the dip coating of a cellulose material, the characterization and the use for oil water separation of this materials.
The topic is introduced well but unfortunately a discussion of the results is missing completely. Furthermore, the description of the experiments does not allow for a reproduction of the experiments. Moreover, there is not even a reference of the separation by the crude membrane in order to verify the efficiency and the use of the coating.
While the topic can be of interest, in the current state I can only recommend a rejection of this manuscript.
Abstract:
Multiple wurds should be in lower case such as energy, field ord contact angle.
Materials:
Please state the purity/grade of the used solvents.
Methods:
The description of the characterization techniques needs to be improved significantly. How have you measured IR? Was it conducted in transmission? How was the sample prepared? How many repetititions and how many scans have been recorded?
For the contact angle measurement, the stabilizing time for the contact angles can be added as well. How often have you repeated the measurements? How have you chosen the spots since your materials seem to be very rough?
In which mode and with which sample preparation have you conducted the FE SEM measurements?
Results:
IR: From which vibration does the band at 700 cm-1 originate?
The bands described in the results seem very plausible. However, can you also discuss these results with other literature? You basically generated a typical reverse-phase chromatography material and therfore multiple comparable studies should exist where similar results have been observed. Can you add literature references to the observed bands and behavior of your material with this coating technique?
The same is true for the further discussion about the FE-SEM results
Figure 3: indicate the color with the respective element and add scale bars in the microscopy images.
Discuss the observed contact angles and the interaction of your membrane with water with literature.
Figure 5: Add the used amount of water in the figure caption. Preferably also the time after the dropping of the water drop and the taking of the picture.
Figure 6: This is rather an experimental or methodical description than a result. You should devinitely describe the procedure and the investigated parameters (amount of water and organic phase) as well as the coloing agent in the method section.
A discussion of the results is completely missing. There is neither a discussion in the results section. nor a standalone discussion section. You should add either one or the other. Without a scientific discussion of the own results, this manuscript is no scientific paper but a report.
Conclusion:
Can you add an outlook or at least how the findings of this manuscript can be applied, either directly or in the future?
Reviewer 2 Report
The authors present a manuscript with the title “Removal of oily contaminants from water by using the hydrophobic nanoparticle incorporated dopamine modified cellulose foam”. The major application the authors suggest for their findings is the removal of oil spills. While this topic is of course important, the work presented here does not appear to be ready for publication yet. Thus, this reviewer suggests to apply some major changes fist.
English language is mostly fine, but the tense used in the main part of the text is either wrong or unusual. This reviewer suggests to request language check by a native speaker.
The introduction is overall to short and could give some actual examples of other materials used for oil water separation. What are the materials/methods actually used for oil spills (state of the art)? Are there bench mark materials?
Section 2.1 is incomplete. The source of chemicals (AgNO3, dopamine, methylene blue) is missing. More details should be given regarding the size/dimensions of the cellulose foams.
Section 2.2 is also not sufficient. Important technical details are missing (e.g. spectral resolution of the IR spectra).
Section 2.3 should be more detailed regarding the chemical background of the modification steps. The formation of the different surface functionalities and modification reagents is insufficient.
Line 164 Magnifications are a number of x-fold (e.g. 1000-fold magnification). The 50/10/5/1 µm are the size of the scale bars, not the magnification.
The overall issue with this manuscript is that it appears to be lacking novelty. Given the suggested application, a large number of publications should be available to compare the results generated here to other studies. Otherwise, this would will not appear to be convincing enough to be considered for publication.
Minor issues:
Line 25 “contact angle” should be “water contact angle”
Line 40/42/48/59/60/62 space missing in between the text and the citation
Line 87/88 both sentences start with “various”, please remove “various” from line 88
Line 100 “on” should be “in”
Line 106 please remove “various”
Line 117 might contain a double space, please check
Line 128 “hydroxyl group has” should “hydroxyl groups have”
Figure 1 IR spectra usually have an inverted x-axis, starting with high values on the left side.
Line 188 “ad” should be “and”
Line 194 “that’s” should “that is”
Line 196 “118 °” remove space in between number and unit
Figure 6 The resolution of the text in the pictures is too low.
Figure 7 Please add error bars to the diagram, y-axis should be “efficiency” instead of “efficiencies”.
Round 2
Reviewer 1 Report
The manuscript has been significantly improved.
Reviewer 2 Report
The manuscript was significantly improved by the authors and can be published in the current form.
A small number of minor issues were found during the second read. These should be addressed before publishing:
Line 110: "cm 1" should be "cm -1"
Line 131: "50 C°" should be "50 °C"
Line 140-142: the empty line should be above the sub-heading, not below
Line 188: "ischaracterized" should be "is characterized"
Line 214: "Green" should be "green"